# Botanical Composition and Diet Quality of the Vicuñas (*Vicugna vicugna* Mol.) in Highland Range of Parinacota, Chile

**DOI:** 10.3390/ani10071205

**Published:** 2020-07-16

**Authors:** Giorgio Castellaro, Carla Orellana, Juan Escanilla, Camilo Bastías, Patrich Cerpa, Luis Raggi

**Affiliations:** 1Faculty of Agriculture Sciences, University of Chile, Santiago 8820808, Chile; carla.orellanam@gmail.com (C.O.); juanescanillacruzat@gmail.com (J.E.); cbastias@hotmail.com (C.B.); patrichcm@gmail.com (P.C); 2Faculty of Veterinary Sciences, Universidad de Chile, Santiago 8820808, Chile; lraggi@uchile.cl

**Keywords:** fecal nitrogen, highland grassland, microhistological technique, nutritional ecology, South American wild camelids

## Abstract

**Simple Summary:**

For the proper management of grazing wild ungulates, it is very important to know the botanical composition and quality of their selected diets. In the case of the vicuña, a wild camelid that lives in the Chilean highlands, there is little information related to these aspects. Therefore, in this work, the variations in the botanical composition and quality of their diets throughout the year were studied, which were estimated by analyzing the plant fragments found in the feces and the concentration of nitrogen within them. The vicuña mainly selects grasses from dry and wet grassland but is also capable of selecting other species, such as graminoids and dicotyledonous herbs. These plants contribute to obtaining a diverse and high-quality diet, this being an efficient foraging strategy to be able to consume a good quality diet, mainly in the months of high nutritional demand, which coincides with the summer rainy season.

**Abstract:**

Understanding the botanical composition of herbivores’ diets and their nutritional quality is an important question in the development of sustainable strategies for the management of natural resources. In Chilean highland vicuña-grazed grasslands, there is little information in this regard and, therefore, this study aimed to determine the year-round profile of the diet’s botanical composition and quality. In highland grasslands, on an area of 21.9 ha, continuously grazed for 3.06 VU/ha/year (18°03′ S, 69°13′ W; 4425 m.o.s.l), twelve feces piles were sampled monthly and were analyzed through microhistology, and the nitrogen concentration [NF, OM basis] was determined. The botanical composition, diversity (J) and selectivity index (E_i_) of the main species were estimated. Diets were composed of dry–grassland grasses (37.7%), wet–grassland grasses (36.6%), graminoids (14.3%) and forbs (10.2%). The diet diversity ranged from 0.79 (dry–winter) to 0.87 (wet–summer). The main dominant grassland species obtained negative E_i_ values. The annual mean value of [NF] was estimated as 1.82%, with a higher value in summer months (2.21%), which coincides with the physiological states of higher nutritional demand. The vicuñas behave like generalist ungulates, having a high degree of selectivity towards grass species, which mostly fulfill a nutritional role in subsistence and a functional role in survival, applying foraging strategies that allow them to obtain a better quality diet during the season of greatest nutritional demand.

## 1. Introduction

Understanding the ecology of herbivores is an important question in the development of strategies focused on providing sustainability to the management of natural resources. The study of the botanical composition of herbivores’ diets represents an important aspect of their ecology and is relevant in the development of management techniques aimed at reducing the impact of animals on vegetation and on the environment [1,2]. By identifying the components of the diet of grazing herbivores, information is generated to determine key aspects of livestock management, including range carrying capacity, choice of grazing sites, estimation of trophic competition with other herbivores, assessment of the nutrient content of the diet, supplementation needs and prediction of the effects of overgrazing [2,3].

The environmental conditions of the high plateau region (Dry Puna) have generated adaptation mechanisms on herbivores of a physiological, morphological and behavioral type. Consequently, the vicuña was identified as a generalist herbivore, but highly selective in the choice of its diet [4,5] and with significant plasticity in its trophic behavior [6]. There is consensus on the relevance of low–stratum grasses within the grassland, especially those belonging to the genera *Deyeuxia, Festuca, Poa* and *Hordeum,* on the contribution to the diet of this camelid, [4,5,7,8,9,10]. Regarding the seasonal variability in the botanical composition of the vicuña diet, Mosca and Puig [5], in the Puna of Salta, Argentina (130 mm annual mean precipitation), observed significant changes in the composition of the diet (e.g., an increase in the intake of grasses during the dry winter period). Borgnia et al. [4], however, described a stable diet composition in the Puna of Catamarca, Argentina (170 mm annual mean precipitation). In alpacas (*Vicugna pacos* Linn.), a domestic South American camelid close to the vicuña, which grazes on annual grasslands in the Chilean central zone with a Mediterranean climate type, important changes in the composition of the diet have been detected between the different grasslands’ phenological periods [11]. Related to the above, the nutritional management of animals under extensive conditions is rather complex, due to the difficulty in nutritionally evaluating the diet consumed, due to the selective behavior of animals and the seasonal and spatial variability in the characteristics of grasslands that are grazed.

Under these conditions, the chemical analysis of the fecal material provides a methodological alternative capable of providing quality and quantity indicators in terms of the food consumed [12,13]. Fecal nitrogen content [NF] has been used in ungulates as an indicator of dietary quality, mainly due to its high correlation with dietary nitrogen content [ND] [14,15,16,17]. In cattle, Holecheck, et al., [18], indicate a correlation coefficient of 0.81 between [NF] and [ND]. Studying various ungulates, Wofford et al., [14] and Aldezabal et al., [19], have determined positive correlations between this indicator and liveweight and population density changes. The use of [NF] as an indicator of dietary quality in wild camelids is rather scarce, with a greater number of antecedents referring to llamas and alpacas [20,21], however Borgnia et al. [22], and Borgnia et al. [4], have provided some values for vicuñas and donkeys in this regard, in conditions of the dry Puna of Argentina. However, under Chilean dry Puna conditions, where meteorological conditions are different, especially in terms of precipitation and thermal amplitude, which determine different botanical compositions and grassland growth patterns, this information is scarce. Based on the aforementioned background, the present study hypothesizes that the vicuña diet is mainly composed of grass species, presenting variations throughout the year in quality (estimated through [NF]), diversity and species composition, with different degrees of selection for the main diet’s species, especially between the dry winter months and the rainy summer months. Therefore, this study aims at determining a profile for the botanical composition of the diet and [NF] of vicuñas throughout one year, under extensive grazing conditions.

## 2. Materials and Methods 

The study was carried out at the captivity vicuñas’ management module, located in Caquena town (18°03′ S, 69°13′ W; 4425 m.o.s.l.), between April 2010 and March 2011. The climate of the sector corresponds to Cold Tundra or Dry Puna (ET) [23]. The mean annual precipitation is 390 mm and is concentrated during the summer period. The mean annual temperature is 2.5 °C, with the warmest months being December and January, as shown in Figure 1.

The soil in the dry–grasslands ecosystems (“pajonal”) are classified as Inceptisols, Cryochrepts [24], characterized by being thin to moderately deep, with medium to coarse textures and with low organic matter content (0.5–2%). The soils of the wet–grassland ecosystem (“bofedal”) are characterized by being organic and hydromorphic, classified as Histosols, Cryofibrist [24], with a soil profile consisting of a mass of herbaceous plant remains in different stages of decomposition [25]. Perennial grasses of the genera *Festuca, Deyeuxia* and *Stipa* are dominant in the dry–grasslands, while in the azonal wet–grasslands ("bofedales"), there are frequent grass species belonging to the genera *Deyeuxia* and *Festuca*, and halophytes, such as the genera *Oxychloe, Distichia, Werneria* and *Carex* [26].

The study was carried out in an area of 21.9 ha, where 43 females with their calves, 14 dry females and 19 males grazed together continuously and without the effects of grazing by other ungulates. These animals represented 67.98 Vicuñas Units (VU) [27], equivalent to 7.45 Standard Animal Units (AU) [2,28], corresponding to a mean stocking rate of 3.06 VU/ha/year/ (0.34 AU/ha/year).

### 2.1. Botanical Composition and Grassland Cover

The evaluations of the botanical composition of the grassland were carried out in two contrasting periods, rainy–summer (January) and dry–winter (July), using the “point transect” method [29]. One hundred points were observed within 20 transects of 50 m length each, which were arranged in the different vegetation elements existing in the experimental area (3 to 4 transects by vegetation units), that were defined in previous studies [26]. Throughout the evaluation period, the animals had free access to all the existing vegetation units in the study area. The relative contribution of each plant species (*CEsp_i_*, %) was calculated by determining the number of hits made on each of the n plant species (*C_i_*) in relation to the total hits made in all grassland plants in each transept.
(1)$$CEsp_{i}=\frac{C_{i}}{\sum_{i=1}^{n}C_{i}}100$$

The vegetation cover calculation was made from all the points where the presence of at least one species was observed. In addition, the presence of mosses, lichens, litter, bare soil, stones and rocks were recorded.

### 2.2. Botanical Composition of the Diet

The botanical composition of the diet was determined by microhistological analysis of feces [2,30,31,32]. The collections of fecal samples were carried out monthly, collecting approximately 50 g of fresh feces, coming from 12 feces piles existing in the experimental area, since the deposit of feces in piles is part of the usual territorial behavior of these camelids, as shown in Figure 2.

Stool samples were dehydrated in a forced air oven at 70 °C for 48 h and ground at 1 mm in a Willey mill. Subsequently, each sample was split into two portions, one for microhistological analysis and the other for nitrogen determination. For microhistology, five slides per sample were prepared, in which 100 visual fields were evaluated under an Olympus optical microscope, model CX21, with a built–in digital camera, using 100X magnification. A valid visual field was considered to be one, which presented at least one identifiable plant fragment [33]. The epidermal fragments present in the feces were identified by comparing what was observed under the microscope with photos and drawings of the reference epidermal patterns obtained for the plants in the area [33]. The plant species were identified when possible but, in some cases, the identification was at the genus or botanical family level. The result of the microscopic reading was expressed as relative frequency, which was transformed into density, using the table’s proposed by Fracker and Brischle (1944) [30,34]. The identified species were grouped into four main functional groups: grasses (Poaceae); graminoids (Cyperaceae and Juncaceae); dicotyledonous herbs; shrub species.

### 2.3. Diet Diversity

Using the data on the diet’s botanical composition, its diversity was determined by calculating the Shannon–Wiener index (*H*):(2)$$H=-\overset{n}{\underset{i=1}{\sum }}P_{i}\;Log_{2}\left(P_{i}\right)$$

The previous index was expressed as relative diversity or equality (*J*) [35,36]:(3)$$J=\frac{H}{Hmax}$$

In the above equations, *P_i_* is the proportion of the species i in the diet and n is the total number of species in the diet. *H_max_* represents the value that H would have if all the species found in the diet had the same frequency ($H_{max}=Log_{2}\left(n\right)$).

### 2.4. Selectivity Index

The Ivlev selectivity index (*E_i_*) was calculated for the main species consumed [35,37], relating the proportion of a species present in the diet (*d_i_*) with its proportion in the grassland (*p_i_*):(4)$$E_{i}=\frac{d_{i}-p_{i}}{d_{i}+p_{i}}$$

The *E_i_* values vary between –1 and 1. Negative values are indicators of rejection towards the species, while positive values indicate preference. Values close to zero reveal indifference to the species in question.

### 2.5. Fecal Nitrogen Content [NF]

This analysis was performed on the fraction of the fecal sample destined for this purpose, using the Kjeldahl method [38]. Values were expressed as percentages of organic dry matter basis.

### 2.6. Experimental Design and Statistical Analysis

The proportion of each species in the diet, diversity and selectivity indices and fecal nitrogen were subjected to an analysis of variance, assuming a completely random design, using a mathematical model of repeated means:*Y_ij_ = µ + Month_i_ + Pile(M)_ij_ + ε_ij_*(5)

The *Y_ij_* is the response variable, *µ* is the general mean, *Month_i_* represents the effect of the i–th month (April, May,…, March), *Pile(M)_ij_* is the effect of the j–th pile nested within the i–th month and *ε_ij_* is experimental error.

Each one of twelve piles from which the fecal samples were extracted were considered repetitions, in turn constituting the experimental unit of the study. Normality in the distribution of the variables studied was analyzed using the Shapiro–Wilk test, with 5% significance. Likewise, the variance was tested for homogeneity by analyzing group means. To detect the differences between the evaluation months, the LSD test at 95% confidence was used. The degree of association between the main functional groups of the species present in the diet and the [NF], was determined through the calculation of the Spearman correlation coefficient [39]. All of the above analyses were performed using Statgraphics Centurion XVI^®^ software.

## 3. Results

### 3.1. Grassland Botanical Composition and Cover

The dominant species in the grasslands was *Festuca orthophylla*, a dry–grassland type grass with a stable contribution that averaged 31.5%. *F. nardifolia* and *Deyeuxia curvula,* wet–grassland type grasses, followed in importance. Among the graminoids, *Oxychloe andina* and *Distichia muscoides* were the most important. The presence of dicotyledonous herbs was low (<5.3%); however, they were more relevant during the rainy–summer period. The vegetation cover was of the order of 70%, but a large part of the bare soil was protected by the presence of litter, as shown in Table 1.

### 3.2. Diet’s Botanical Composition

The most important species in the diet of vicuñas were grasses, with similar annual mean percentages between those of the dry–grassland environment (37.7 ± 13.1%) and the wet–grassland (36.5 ± 8.1%), although with greater variability in the first group (variation coefficient of 34.87% vs. 22.2%, respectively). However, significant changes between the months (F ratio = 51.15 and 22.81, for dry and wet grasses, respectively; *p* ≤ 0.001; degree of freedom =11) in the contribution of both groups were evident. In this regard, the contribution of wet–grassland grasses was higher during the dry months (April–September), the opposite occurring with the dry–grassland grasses, as shown in Figure 3.

*Deyeuxia deserticola* was important within the group of dry–grassland grasses, with an annual mean contribution of 22.0 ± 5.1%, with relatively stable values throughout the year, except in the months of October, November and December, where a greater contribution was evident. The dominant species of dry–grassland, *F. orthophylla*, contributed 8.3% to the diet, while *D. heterophylla* obtained values of the order of 6.4%, as shown in Table 2. The percentage of these latter species tends to increase from the dry–winter season to the spring and summer months. It should be noted that both *D. deserticola* and *D. heterophylla* are species that were not detected in the grassland’s botanical composition, but still contributed to the vicuña diet, suggesting a high selectivity index for the species.

*Deschampsia caespitosa* and *D. curvula* were important in the group of wet–grassland grasses, with average annual contributions of 15.4 ± 5.9% and 11.2 ± 8.5%, respectively. *D. caespitosa* contributed in a lesser proportion during the dry season, tending to increase its participation in the diet at the beginning of the rainy season and, as was the case with *D. deserticola*, it would be a highly selected species, since, although its participation in the grassland could not be detected, it appeared in a high percentage in the diet. The opposite occurred with *D. curvula*, whose contribution was high in the dry season and higher than that recorded in the grassland, while in the spring and summer months, their contribution percentages were lower than those offered by the grassland, as shown in Table 2.

The third group of importance were graminoids, with an annual average of 14.3 ± 6.0% and a relatively stable contribution to the vicuña diet throughout the year. Within this group of species, *Oxychloe andina*, *Distichia muscoides* and *Carex incurva* were the most important, but with percentages that did not exceed 7% on average and with values frequently lower than those offered by the grassland, as shown in Table 2. The observed trend in the intake of *O. andina* suggests a higher consumption during the dry season, unlike what was observed in the case of *C. incurva*, where the greatest contribution to the diet was during the spring and summer rainy months.

Dicotyledonous herbs averaged a contribution of 10.2 ± 4.6% and with a higher proportion during the dry winter period, as shown in Table 2. Within this group, *Gentiana prostrata* and *Lilaeopsis andina* were important, but with percentages that averaged between 3 and 4%. The contribution of woody species was very low, with an annual average of 1.3 ± 1.8%, especially during the winter months. *Parastrephia lucida*, with almost no presence in the grassland, was the most least relevant species in this group, as shown in Table 2.

### 3.3. Diet’s Relative Diversity Index (J)

The number of plant species identified in the diet ranged from 12 to 17, depending on the time of year. Regarding the diet’s relative diversity index, significant variations were observed between months (*p* ≤ 0.05), with a minimum value of 0.79 ± 0.03 in July, during the dry–winter season (April to November) and 0.87 ± 0.03 in December, during the wet–summer season (December to March), as shown in Figure 4.

### 3.4. Selectivity of the Main Consumed Species (E_i_, Ivlev’s Index) 

In the selectivity results, it is important to emphasize the behavior of those dominant species in the botanical composition of the diet, whose contribution in the grasslands could not be quantified. This is the case for the dry–grassland grasses, *D. deserticola* and *D. heterophylla*, which probably present a high degree of selection (E_i_ ≈ 1), since despite being practically undetectable in the grassland, they contributed with a high percentage in the diet, especially in the wet–summer season. A similar situation was observed in the wet–grassland grasses, *D. caespitosa, F. nardifolia* and *D. crhysantha*, as well as in some graminoids and in most dicotyledonous herbs. 

The grass species, *F. orthophylla* and *D. curvula*, always rejected by the vicuñas, similar to graminoids, such as *O. andina* and *D. muscoides,* observed negative selectivity values throughout the year, while *C. incurva* was selected from September to December, but the rest of the year was rejected, as shown in Table 3.

### 3.5. Fecal Nitrogen [NF]

The annual mean [NF] was 1.83 ± 0.3% (organic–matter basis), however, significant differences (*p* ≤ 0.05) were observed between months, with a tendency to obtain higher values during the rainy–summer season, especially between December and March, as shown in Figure 5.

The group of dry–grassland grasses was the only functional group that was positively and significantly correlated with [NF], while in the rest of the functional groups, although significant correlations were detected, these were low and negative, as shown in Table 4.

## 4. Discussion

### 4.1. Grassland’s Botanical Composition and Cover

The detected changes in the grassland cover and botanical composition agrees with what was found in other studies [26], being determined by the degree of hydromorphism and soil salinity in relation to the sensitivity of plant species to these edaphic factors. The variations in the contribution of the different species between the evaluated months could be attributed to environmental variations associated with the thermo–pluviometric regime and water availability in the soils. These aspects are important since they not only determine the quantity but also the nutritional quality of the forage offered. In this regard, studies carried out by Castellaro and Araya [26] indicate different contributions of crude protein and metabolizable energy for different types of grassland, which vary depending on the time of year. In general, wet grasslands show low variation in the metabolizable energy concentration (7.1–7.7 MJ/kg DM), with the crude protein content being more variable (6.8–11.5%). The latter is associated with the summer regrowth of perennials plants and a greater contribution of dicotyledonous herbs during that season, which, given their higher protein content, would contribute to raising the grassland protein concentration [40]. In the case of dry–grassland grasses, where *F. orthofhylla* dominates, the same authors report values of metabolizable energy between 3.3 and 5.4 MJ/kg DM (and in some cases up to 7.4 MJ/kg) and crude protein percentages between 0.9 and 3.4%, between the dry and growing–wet season; however, these variations are less marked.

### 4.2. Diet’s Botanical Composition

The contribution of grass species has been considered by various authors as the main component of the vicuña diet [7,8], representing around 70% of the total consumed species [4,5]. This coincides with what was obtained in this work but differs from results presented by Tirado et al. [41], who report percentages of grasses of around 40% in vicuña diets in the Atacama Highland grassland. This could be attributed to important differences in the type of natural grasslands, since in our experimental conditions, the environment is more humid and the presence of shrub species was very low, not even being detected in the composition of the grasslands, as shown in Table 1.

Regarding the content of dry–grassland grasses, in alpaca diets, Castellaro et al. [42], report an increase from 17.1% in summer to 26.2% during winter. The same authors point out a significant proportion of *F. orthophylla* in the diet (i.e., 13.5 ± 3.8% in winter and 5.6 ± 3.0% in summer). However, the increase observed in the contribution of *F. orthophylla* and *D. heterophylla* during the spring and summer months, could probably be due to the fact that, during this last period, more tender tillers, the product of lower cell wall contents and higher water percentages, occur in these grasses, which are consumed in a greater proportion by grazing animals. 

Castellaro et al. [42] describe important proportions of wet–grassland grasses in the alpaca diet, reaching 50.9% of the dry–winter diet and 58.5% of the diet during the wet–summer period. These values are higher than those found in the present study, but similar to those determined by Miranda et al. [10], who studied the diet of juvenile vicuñas in the grassland of the Chilean highlands located near the place where this study was conducted. These authors determined a participation of *D. curvula* of 20.19%, and similar values were found in this study during the dry–winter period, as shown in Table 2. However, during the summer season, the percentages of this species were lower than those determined in the study by Miranda et al. [10], who reported 26.8% during the wet–summer season. The probable difference in the abundance of *D. curvula* in the grasslands, could induce a different selective behavior on the part of the animals.

The grass *F. nadifolia* was another species that was consumed in a relatively high proportion during the summer months (4.6 to 14%). However, its contribution decreased significantly during the winter (<5%). This could be attributed to changes in its nutritional quality, determined between the summer–growth and winter–dormant period [26]. However, its dietary contribution was always lower than its relative appearance in grasslands, which is typical for "less desirable" species that tends to be rejected [2]. Other grasses, such as *D. chrysantha*, observed a high consumption during the dry months (April to July). This species is reported as a high–quality grass, which it maintains throughout the year [26], however its presence in grasslands is scarce, making it a “desirable” and selected species [2], an aspect that is discussed later. Furthermore, this species grows in flooded soil conditions, a situation that is accentuated during the wet–summer months and a situation that may restrict its accessibility to grazing.

*O. andina*, has been described by other authors as an important component in the diet of vicuñas [4,5,8,10] and of alpacas and llamas [42], with a dietary contribution between 11 and 17%, especially during the dry–winter season. This coincides with what was found in our study, where, in total, the graminoids contributed with a mean of 14% to the vicuña diet, and *O. andina* was the major component of this group, contributing up to 11.6% in June. Dicotyledonous herbs have been described as an important component of the vicuña diet in the wet–summer period [5,7,8]. However, Borgnia et al. [5] found very low proportions of herbs in the vicuña diet, while Aguilar and Neumann [9] did not identify this group in the diet of this camelid, which coincided with our results. In alpacas, Castellaro et al. [42] determined 0.9% of herbaceous species in the dry–winter diet, increasing to 1.2% in the wet–summer period, coinciding with the trends found by Farfán and Bryant [43] in the diet of this camelid in natural grasslands of the Peruvian Puna. These results are contrasted with those found in our work, where dicotyledonous herbs represented an annual average of 10.2%, especially at the beginning of the dry–winter season (April), as shown in Figure 1 and Table 2. Within this group, *Gentiana prostrata*, as shown in Table 2, was important—a species that has low fiber contents and a high concentration of metabolizable energy and crude protein [26]. Being a relatively scarce species in the wet–grassland botanical composition, and contributing significantly to the vicuñas winter diet, it could be considered a “desirable” species that contributes to improving the nutritional quality of the diet during this period.

Koford [7], does not consider the shrub intake as a habitual trophic behavior in vicuña, agreeing with other authors who mention that the intake of shrub species is rare in camelids [6,8,44,45]. Castellaro et al. [42] (2004) report a content of shrub species close to 6% in the alpaca diet throughout the year. Borgnia et al. [4] and Mosca and Puig [5], have also found a higher proportion of shrubs in the vicuña diet (10 to 17%), although under conditions that differ from the present study in terms of the availability and diversity of this group of plants. Even higher proportions are reported in a study by Tirado et al. [41], who point out a contribution of shrub species of almost 34% in the summer diet of vicuñas in prairies of the Atacama highland ranges. Benítez et al. [46] emphasize the impact of the shrub species of the steppes, on the quality of the vicuña diet, considering this group as an important resource in the last period of the wet–season, at which time the shrubs show low lignified twigs with high protein content.

### 4.3. Diet’s Relative Diversity Index (J)

The values of the relative diversity index (J) obtained in this study (0.79–0.87) differ from that found by Borgnia et al. [4], who determined a lower value (0.7 ± 0.1) but with a greater number of species (n = 23 to 25), finding no differences between the rainy and dry–seasons. Mosca and Puig [5], in a locality of the Argentine Puna, registered values of J of 0.86, but with a greater participation of the species (n = 25). Castellaro et al. [42], in alpaca diets, found average values of 0.81 in the wet–summer season while, during the dry–winter season, this value increased to 0.85 as a result of a more equitable consumption of plant species during this period. The results obtained in the present study suggest a “generalism” cohesion in the dietary habits of vicuña, since a balance was observed in the relative contribution of most of the species that were consumed and that were potentially available in the grasslands. This trophic behavior is consistent with that described by Hanley [47] for generalist ungulates, coinciding with that indicated by Borgnia et al. [4] and Mosca and Puig [5], who, in addition to this classification, define vicuñas as ungulates that consume a diet with a high percentage of grasses.

### 4.4. Selectivity of the Main Consumed Species (Ivlev’s Index)

Certain dry–grassland grasses (*D. desertícola* and *D. heterophylla*), some wet–grassland grasses (*D. caespitosa*, *F. nardifolia* and *D. crhysantha*) and most dicotyledonous herbs, showed a high degree of selection, although they contributed in low proportions to grassland dry matter. Therefore, they seem to fulfill a nutritional production role and a functional role as diet-enhancing species [37], which could be attributed to the relatively high crude protein contents in their tissues [26] and the low content of secondary compounds that could affect the intake [48]. A high contribution to the grassland botanical composition of species, such as those mentioned above, would be an indicator of the “excellent to good range condition” and should be prioritized when carrying out restoration work on degraded grasslands.

Unlike the above, *F. orthophylla*—the most abundant dry–grassland species—is negatively selected, especially during dry winter months. This grass would be a forced selectivity species, fulfilling a nutritional role in subsistence and a functional role in survival [37]. This could be attributed to its low contributions of crude protein and high content of fiber in their tissues [26]. In contrast to our results, Borgnia et al. [4] point out that *F. orthophylla* is consumed by vicuñas in proportion to its availability.

In the case of *D. curvula*, a lower degree of rejection was observed, averaging an Ivlev’s index that classifies it as a species of proportional selectivity, being consumed according to its availability in the grassland, fulfilling a nutritional maintenance role and a functional role of volume [37]. The same can be noted for the graminoids *C. incurva* and *D. muscoides*. In the case of *O. andina*, it would be a species of forced selectivity, especially in certain specific situations, such as when the grazing animal cannot separate it from other selected plants because it has certain consumable tissues only at a certain time of the year or when there is a limited supply of desirable and preferred vegetable species in the grassland [49]. *O. andina* presents leaves and stems of coriaceous consistency, sharp-pointed and of low height, which probably constitute a limitation for its consumption. The values obtained in the Ivlev’s index for this species coincide with those obtained by Castellaro et al. [42], who consider it as an “undesirable” species in grasslands destined for llama and alpaca grazing.

In highland wet–grassland ecosystems, Orellana et al. [50] indicate that, depending on the herbivore, graminoids can be considered as species of forced intake and/or of proportional intake to the abundance in the grassland, associating them with a nutritional role of subsistence and maintenance, respectively.

According to the optimal foraging theory proposed by Stephens and Krebs [51], the grazing strategy of the vicuñas studied would correspond to the maximization of nutrient intake per unit of time, which is reflected in their ability to select the trophic items with the highest nutritional value, within the possibilities offered by their environment. It should be noted that most of the Ei calculated for the aforementioned plant species do not correlate with their respective relative availability in the grassland, as shown in Figure 6, so the selection of these species would probably be determined by their nutritional quality rather than by their availability [52]. 

Such a pattern of behavioral adaptation is closely related to other types of morpho–physiological adaptations present in the vicuña, such as greater lip mobility, the presence of two–toed feet with toenails and soft foot pads and greater digestive and respiratory efficiency, which allow this camelid the viability of herbivorous life in conditions of scarce nutritional resources.

### 4.5. Fecal Nitrogen [NF]

Our results show a mean [NF] of 2.12% during the wet months (December–March), while in the dry months (April–November), the mean [NF] was 1.68%, as shown in Figure 5. In studies carried out in the Argentine Puna [4,22], variations in the [NF] have been found between the different seasons of the year, decreasing during the dry–winter months, observing values of 1.48 % in March, 1.26% in May, 0.96% in September and 1.23% in October. Working in a similar environment, Benítez et al. [46] indicate an annual mean of 1.26% in [NF]. These values are lower than those presented in this work, probably due to the lower proportion of species with high protein content in the diet consumed by these populations of vicuñas. However, the trends in the variability of the fecal indicator were similar. Kamler and Homolka [53], working with red deer, describe similar values to those presented here, also observing a decrease in [NF] during winter. The decrease in [NF] during unfavorable periods has also been mentioned by other authors in works referring to different herbivores [13,15,54,55]. Higher [NF] contents are correlated with a diet with a higher protein content, lower fiber and lignin content and higher digestibility, both of the dry matter and of the protein consumed [13,56,57]. The improvement in the quality of the ingested diet in summer could be due to the greater proportion of grasses in the dry–grassland environment (“pajonal”) in immature phenological states and with high protein content in their tissues, such as *D. deserticola*, for which crude protein values of almost 18% have been determined [26]. The best dietary quality, based on the highest concentration of [NF], is found during the summer–rainy months (2.13% between December and March). This is of high nutritional importance, given that, during this time, vicuñas go through the last third of gestation and the first weeks of the lactation stage, in which the highest nutritional requirements are present [58,59,60].

## 5. Conclusions

Under the conditions in which this work was carried out, it can be concluded that the vicuñas living in the highlands of Parinacota, behave as generalist ungulates, with a high degree of selectivity towards grass species. Within this functional group, the most abundant species in the grasslands evaluated fulfill a nutritional role in subsistence and a functional role in survival. However, other grasses and graminoids carry out a nutritional role in maintenance and a functional role in volume. The vicuña presents important changes in the intake of different functional groups of plants throughout the year, through the use of foraging strategies that allow it to find and select grass species with high nutritional value, which have low presence in grasslands. In this way, vicuñas obtain a better-quality diet which allows them to satisfy their nutritional requirements, especially during the season of greatest demand (summer). This can be corroborated with the highest concentrations of [NF] that were determined during this period.

## Figures and Tables

**Figure 1 animals-10-01205-f001:**
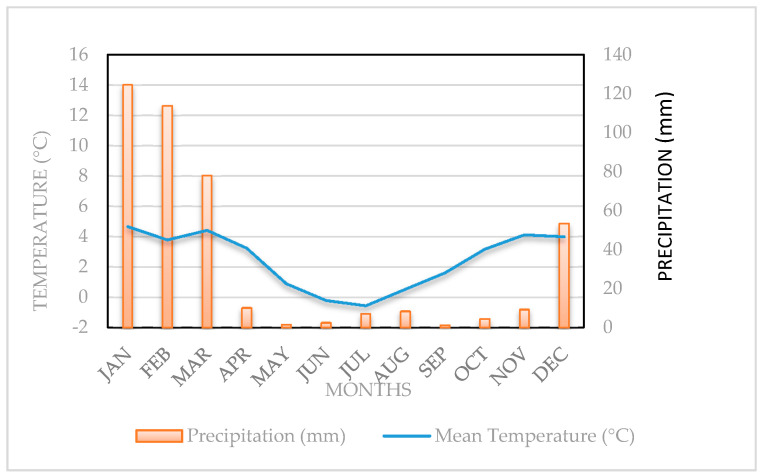
Ombrothermic diagram of Caquena, Putre County, Parinacota, Chile.

**Figure 2 animals-10-01205-f002:**
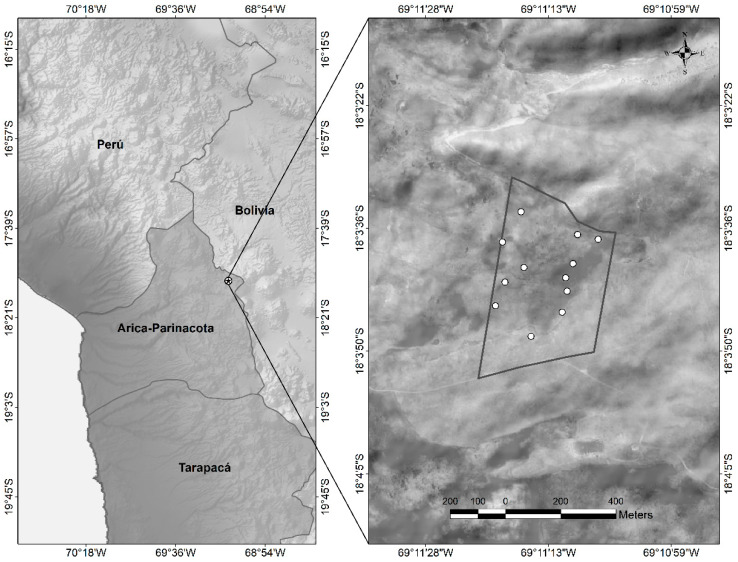
Location of the experimental area and distribution of the feces piles within the area grazed by vicuñas.

**Figure 3 animals-10-01205-f003:**
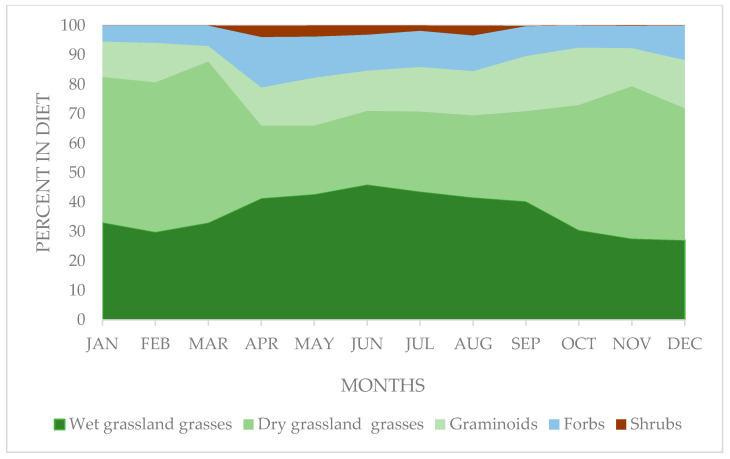
Monthly variation in the contribution of the main functional groups of plants in the vicuña diet in natural grasslands of a sector of the Chilean highlands.

**Figure 4 animals-10-01205-f004:**
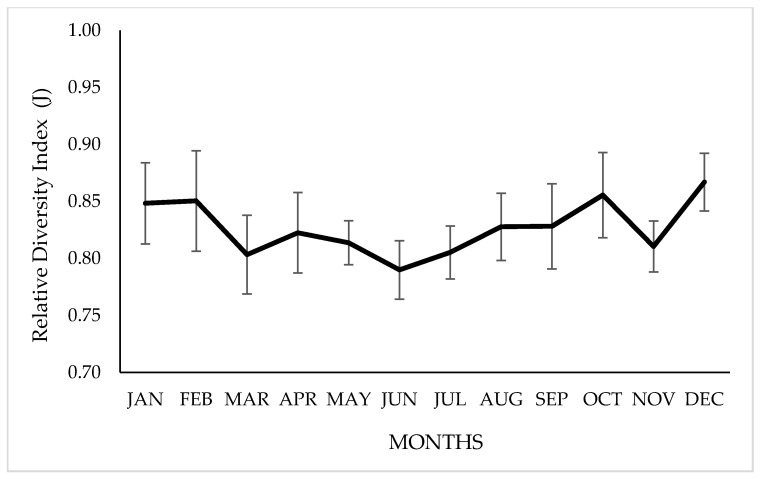
Monthly variation in the relative diversity index (J) of the vicuña diet, grazing on natural grasslands of Chilean highlands. Bars indicate a standard deviation above or below the means.

**Figure 5 animals-10-01205-f005:**
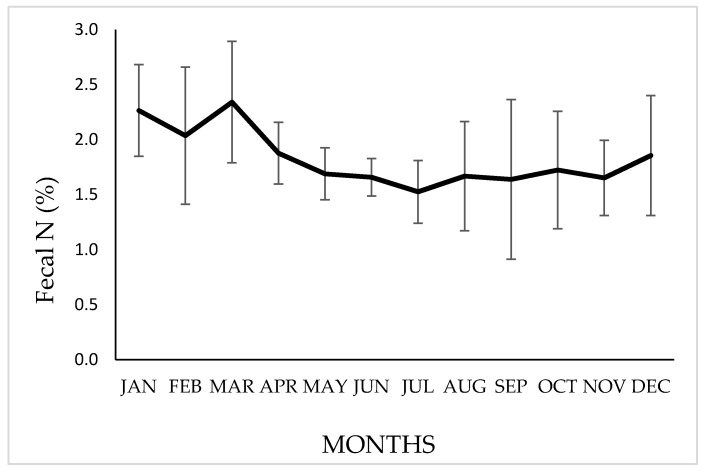
Variation in the N percentage in feces (organic–matter basis) of vicuñas grazing on natural grasslands of a Chilean highlands. Bars indicate a standard deviation above or below the means.

**Figure 6 animals-10-01205-f006:**
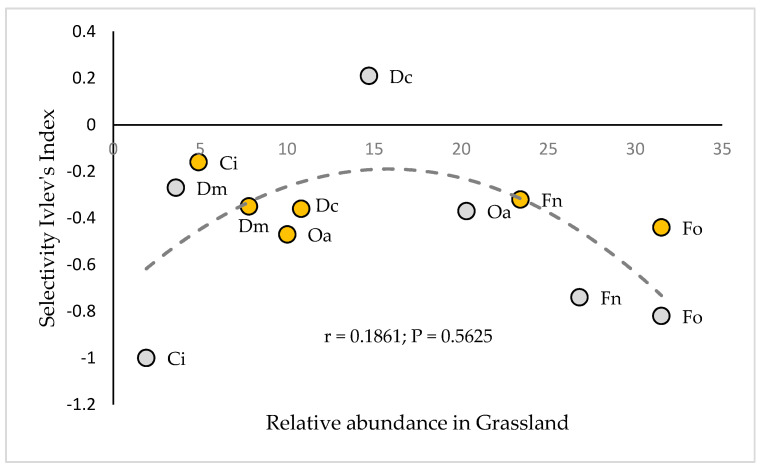
Relationship between the relative abundance of the species in the prairie and its corresponding selectivity index, measured in the months of January (wet–summer season) and July (dry–winter season). Fo = *Festuca orthophylla*; Dc = *Deyeuxia curvula*; Fn = *Festuca nardifolia*; Ci = *Carex incurva*; Oa = *Oxychloe andina*; Dm = *Distichia muscoides*. Orange circles correspond to January measurements and gray circles correspond to July measurements.

**Table 1 animals-10-01205-t001:** Botanical composition of the grasslands where the study was carried out, in two contrasting periods: January (rainy–summer) and July (dry–winter) in Caquena, Putre County (18°03’ S, 69°13’ W; 4425 m.o.s.l.), Arica and Parinacota Region, Chile.

Plant Species	January	July
Grasses		
*Festuca orthophylla* Pilg.	31.5	31.5
*Festuca nardifolia* Griseb.	23.4	26.8
*Deyeuxia curvula* Wedd.	10.8	14.7
*Deyeuxia jamesonii* (Steud.) Munro ex Wedd.	0.4	0.0
Graminoids		
*Oxychloe andina* Phil.	10.0	20.3
*Disticha muscoides* Nees and Meyen	7.8	3.6
*Carex incurva* Lightf.	4.9	1.9
*Eleocharis sp.*	0.7	0.0
Dicotyledonous herbs		
*Werneria pygmaea* Gillies	5.3	0.0
*Lachemilla pinnata* (Ruiz and Pav.) Rothm.	1.2	0.0
*Arenaria rivularis* Phil.	0.4	0.4
*Hypochaeris taraxacoides* (Walp.) Benth.	2.7	0.7
*Plantago* sp.	0.8	0.0
Total	100.0	100.0
Ground Cover	71.6	69.0
Mosses and lichens	0.0	0.0
Bare soil	13.2	11.0
Stones	4.1	4.1
Rocks	1.3	1.3
Litter	9.9	14.7

**Table 2 animals-10-01205-t002:** Mean monthly contribution of different plant species to the diet of vicuñas that graze on natural grasslands of a sector of the Chilean highlands in Caquena, Putre County (18°03 S, 69°13’ W; 4425 m.o.s. l.), Arica and Parinacota Region, Chile.

Plant Species	Jan.	Feb.	Mar.	Apr.	May	Jun.	Jul.	Aug.	Sep.	Oct.	Nov.	Dec.	Annual Mean
Wet–grassland grasses													
*Deschampsia caespitosa*	14.90	12.15	20.33	10.86	11.34	13.87	10.93	13.31	25.44	14.71	20.48	16.60	15.41
*Deyeuxia curvula*	5.55	3.17	3.06	17.83	17.58	24.59	22.73	20.00	6.26	4.92	3.26	5.44	11.20
*Festuca nardifolia*	11.93	14.18	9.58	3.05	3.21	3.44	3.99	4.50	4.86	10.41	3.62	4.61	6.45
*Deyeuxia chrysantha*	0.56	0.31	0.16	9.59	10.65	4.17	6.02	3.88	3.01	0.50	0.26	0.51	3.30
*Agrostis tolucencis*	0.26	0.04	0.00	0.09	0.00	0.00	0.00	0.00	0.51	0.05	0.06	0.00	0.08
*Deyeuxia jamesonii*	0.00	0.09	0.00	0.00	0.00	0.00	0.00	0.00	0.28	0.00	0.00	0.00	0.03
Sub–total	33.2	29.9	33.1	41.4	42.8	46.1	43.7	41.7	40.4	30.6	27.7	27.2	36.5
Dry –grassland grasses													
*Deyeuxia deserticola*	21.05	23.76	23.69	20.14	20.11	20.64	22.05	19.79	15.03	25.12	29.20	23.31	21.99
*Festuca ortophylla*	12.46	13.30	18.36	0.90	1.16	2.86	3.16	5.47	8.12	8.97	12.88	11.41	8.25
*Deyeuxia heterophylla*	14.86	12.13	10.26	2.48	1.06	0.38	1.65	1.70	6.19	7.88	9.31	9.15	6.42
*Deyeuxia antoniana*	0.00	0.00	0.00	0.48	0.14	0.18	0.00	0.58	0.33	0.05	0.00	0.00	0.15
*Deyeuxia breviaristata*	0.00	0.06	0.04	0.62	0.77	0.93	0.31	0.32	0.17	0.00	0.00	0.00	0.27
*Poa lilloi*	0.58	1.50	2.25	0.00	0.00	0.00	0.00	0.00	0.28	0.00	0.21	0.44	0.44
*Stipa leptostachia*	0.41	0.08	0.09	0.00	0.00	0.00	0.00	0.00	0.45	0.43	0.17	0.44	0.17
Sub–total	49.4	50.8	54.7	24.6	23.2	25.0	27.2	27.9	30.6	42.5	51.8	44.7	37.7
Graminoids													
*Oxychloe andina*	3.81	3.58	1.61	7.79	10.06	11.58	10.24	11.60	7.13	4.81	5.24	3.56	6.75
*Distichia muscoides*	4.10	4.52	1.10	3.69	2.58	0.96	3.32	2.65	7.24	4.68	2.99	7.06	3.74
*Carex incurvula*	4.12	5.25	2.54	0.00	0.00	0.00	0.00	0.06	3.86	9.32	4.57	5.77	2.96
*Scirpus* sp.	0.00	0.00	0.00	0.54	1.18	0.48	0.70	0.13	0.44	0.60	0.12	0.05	0.35
*Juncus* sp.	0.00	0.00	0.00	0.50	1.69	0.12	0.53	0.18	0.09	0.11	0.00	0.00	0.27
*Eleocharis pseudoalbibracteata*	0.00	0.00	0.00	0.48	0.80	0.50	0.34	0.41	0.00	0.00	0.00	0.00	0.21
Sub–total	12.0	13.4	5.2	13.0	16.3	13.6	15.1	15.0	18.8	19.5	12.9	16.4	14.3
Dicotyledonous herbs													
*Gentiana prostrata*	0.00	0.00	0.00	10.81	9.35	7.71	9.30	9.65	0.24	0.00	0.05	0.00	3.92
*Lilaeopsis andina*	3.09	5.12	5.96	0.00	0.00	0.00	0.00	0.00	7.00	5.30	4.39	7.27	3.18
*Aa nervosa*	0.00	0.00	0.00	5.09	2.86	2.19	2.26	2.21	0.11	0.48	0.10	0.05	1.28
*Cotula mexicana*	1.50	0.05	0.12	0.05	0.00	0.00	0.00	0.00	0.94	0.87	2.33	2.91	0.73
*Miriophyllum acuaticum*	0.22	0.00	0.00	0.87	1.39	1.85	0.62	0.19	0.35	0.20	0.00	0.32	0.50
*Ranunculus uniflorus*	0.11	0.68	0.74	0.00	0.00	0.00	0.00	0.00	0.56	0.43	0.61	1.06	0.35
*Hypochaeris etchegarai*	0.00	0.00	0.04	0.31	0.33	0.47	0.09	0.06	0.17	0.00	0.00	0.00	0.12
*Pratia repens*	0.45	0.00	0.00	0.00	0.00	0.00	0.00	0.00	0.25	0.00	0.00	0.00	0.06
*Alchemilla diplophylla*	0.05	0.04	0.00	0.00	0.00	0.00	0.00	0.00	0.30	0.00	0.00	0.00	0.03
*Plantago barbata*	0.00	0.00	0.00	0.00	0.00	0.00	0.00	0.00	0.12	0.05	0.00	0.05	0.02
*Werneria pygmaea*	0.00	0.00	0.00	0.00	0.00	0.00	0.00	0.00	0.00	0.10	0.05	0.00	0.01
*Pacezia pygmaea*	0.00	0.00	0.08	0.00	0.00	0.00	0.00	0.00	0.00	0.00	0.00	0.00	0.01
*Alchemilla pinnata*	0.00	0.00	0.00	0.00	0.00	0.00	0.00	0.00	0.10	0.00	0.00	0.00	0.01
*Astragalus* sp.	0.00	0.00	0.00	0.00	0.00	0.00	0.00	0.00	0.08	0.00	0.00	0.00	0.01
Sub–total	5.4	5.9	6.9	17.1	13.9	12.2	12.3	12.1	10.1	7.4	7.5	11.7	10.2
Woody species													
*Parastrephia lucida*	0.00	0.00	0.00	3.86	3.73	3.08	1.77	3.32	0.08	0.00	0.10	0.00	1.33

**Table 3 animals-10-01205-t003:** Ivlev’s selectivity index (monthly averages ± standard deviation) for relevant species in the vicuña diet, which have important participation in the botanical composition of the grassland ^1^.

	Plant Species				
Month	*F. orthophylla*	*D. curvula*	*C. incurva*	*O. andina*	*D. muscoides*
January	–0.44 ± 0.09	–0.36 ± 0.23	–0.16 ± 0.30	–0.47 ± 0.18	–0.35 ± 0.26
February	–0.41 ± 0.11	–0.57 ± 0.22	–0.10 ± 0.43	–0.50 ± 0.22	–0.34 ± 0.32
March	–0.27 ± 0.10	–0.50 ± 0.24	–0.35 ± 0.21	–0.74 ± 0.16	–0.78 ± 0.19
April	–0.95 ± 0.05	0.09 ± 0.08	–1.00	–0.46 ± 0.13	–0.20 ± 0.54
May	–0.93 ± 0.04	0.08 ± 0.07	–1.00	–0.35 ± 0.16	–0.33 ± 0.48
June	–0.84 ± 0.07	0.25 ± 0.03	–1.00	–0.28 ± 0.10	–0.71 ± 0.40
July	–0.82 ± 0.06	0.21 ± 0.05	–1.00	–0.35 ± 0.17	–0.27 ± 0.54
August	–0.71 ± 0.08	0.15 ± 0.07	–0.95 ± 0.16	–0.28 ± 0.09	–0.28 ± 0.46
September	–0.61 ± 0.19	–0.43 ± 0.22	0.21 ± 0.43	–0.49 ± 0.12	0.22 ± 0.31
October	–0.56 ± 0.11	–0.51 ± 0.14	0.49 ± 0.50	–0.63 ± 0.14	–0.03 ± 0.49
November	–0.43 ± 0.11	–0.65 ± 0.15	0.38 ± 0.14	–0.61 ± 0.18	–0.13 ± 0.23
December	–0.49 ± 0.22	–0.36 ± 0.20	0.03 ± 0.22	–0.50 ± 0.21	–0.10 ± 0.25
Mean	–0.62 ± 0.24	–0.22 ± 0.37	–0.37 ± 0.62	–0.47 ± 0.20	–0.28 ± 0.46

^1^ The Ei values vary between –1 and 1. Negative values are indicators of rejection towards the species, while positive values indicate preference. Values close to zero indicate an indifference towards the plant species by the herbivore [35,37].

**Table 4 animals-10-01205-t004:** Spearman’s correlation coefficients between the functional groups of plants in the vicuña diet and the N content of their feces.

	Woody Species	Wet–Grassland Grasses	Dry–Grassland Grasses	Graminoids	Herbs
**N Fecal Nitrogen**	–0.2768	–0.3555	0.4031	–0.2434	–0.2231
**n**	144	144	144	144	144
***p*-Value**	0.0009	0.0001	0.0001	0.0036	0.0076

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
