# Peer review of "Botanical Composition and Diet Quality of the Vicuñas (*Vicugna vicugna* Mol.) in Highland Range of Parinacota, Chile"

_animals, 2020, doi:10.3390/ani10071205_

Round 1

Reviewer 1 Report

Although many studies have investigated the ecology of herbivorous vertebrates there are still substations knowledge gaps, namely, regarding the prey selection dynamics and the subjacent environmental factors. As such, analysing a single species inhabiting a simple ecosystem under harsh, highly seasonal conditions provides an excellent model. The manuscript is well-written and supported by relevant literature. Field and lab methods and correct and statistics are mostly sound. Beside some minor suggestions I am adding some recommendation for a better exploitation of the dataset. Overall, I am convinced that authors will be perfectly able to include the in a revised version of the work.

Detailed comments:

Introduction

Line 81

In which sense the conditions of the Chilean puna are different? Less water availability? Higher thermal amplitude? Different seasonality?

Line 86

Which is the hypothesis regarding the seasonality? Should vicuñas simply mirror the plant availability or would be more selective in the dry season?

Material and methods

Line 113

Was the area free of lifestock? Was guanaco absent? Was water available or does it constrain the vicuña ecology?

Line 118

Replace “point transept” by “point transect”.

Line 177

You can calculate confidence intervals of the Ivlev’s index after Strauss (1979).

Strauss, R.E. (1979). Reliability estimates of Ivlev’s electivity index, the forage ratio, and a proposed linear index of food selection . T. Am. Fish. Soc . 111 , 517 – 522 .

Results

Line 226

Indicate the value of the statistic(s) and the degrees of freedom.

Line 289

Replace “desertícola” by “deserticola”.

Line 296

This dataset is excellent to test for correlation between selection and abundance in the environment of the main plant species according the threee models of Stamp et al. (1981):

1) Prey switching: animals select prey items to minimize time; selection is expected to be positively

correlated with abundance

2) Energy optimization: animals select the most valuable prey in terms of energetic content; selection and

abundance should be at least not negatively correlated.

3) Nutrient optimization: animals select items to fulfil their nutrient (or water) requirements or to prevent the accumulation of secondary compounds, selection and abundance are negatively

Stamps , J. , Tanaka , S. & Krishnan , V.V. (1981) Th e relationship between selectivity and food abundance in a juvenile lizard . Ecology , 64 , 1079 - 1092 .

Table 3

Replace “Vegetal species” by “Plant species”

Discussion

Line 252

I wonder is water content matters here.

Line 396

Considering the seasonal variation of plant diversity due the phenology of different species this already suggest diet selection

Line 407

I think authors should also consider the water content and the presence of secondary compounds.

Tirado, C.; Cortés, A.; Miranda-Urbina, E. & Carretero M. A. (2012): Trophic preferences in an assemblage of mammal herbivores from Andean Puna (Northern Chile). Journal of Arid Environments, 79: 8-12.

Line 430

Authors can test if vicuñas are optimizing energy, time or nutrients but correlating selectivity and abundance. See above.

Line 468

The importance of the nutritional value is pending on the statistics.

Author Response

Dear Reviewer

Most of the changes suggested, were made in the new manuscript and are highlighted in yellow with red text.

Reviewer 2 Report

The manuscript entitled "Botanical composition and diet’s quality of the 2 vicuñas (Vicugna vicugna Mol.) in highland range of 3 Parinacota, Chile" presents additional information on the diet of vicunas that is worth publishing. The methods are appropriate, although the description can be improved in some instances. However, I have some concerns about the presentation and discussion of the results that should be improved. Detailed comments and corrections are provided in the PDF, but this is my major point:

The authors set out in the introduction to provide important information that is relevant for "the development of management techniques aimed at reducing the impact of animals on vegetation and on the environment" and to "determine key aspects of livestock management, including range carrying capacity, choice of grazing sites, estimation of trophic competition with other herbivores, assessment of the nutrient content of the diet, supplementation needs and prediction of the effects of overgrazing." Unfortunately, nothing of this is taken up later in the discussion, which is mostly only a comparison of the results of this study with previous studies on vicuna and alpaca. The authors should put forward some ideas that explain differences, and tell the reader why this is relevant for the management of the species. Right now, the reader is mostly left alone with the interpretation of the results.

I think this paper can be improved considerably with some efford, and requires additional checking for English language. I am happy to review a revised version of the manuscript.

Author Response

Dear Reviewer 2

Most of the changes suggested, 2 were made in the new manuscript and are highlighted in green with red text.

Round 2

Reviewer 2 Report

The manuscript has been improved considerably, only few minor points remain that should be addressed (see attached PDF).

Author Response

Dear Reviewer ...

Thank you very much again for your detailed review.
All changes suggested by you in the draft were considered and are highlighted in green and red in the new manuscript that I attach.

Sincerely
